

# Dynamics of green and blue water flows and their controlling factors in Heihe River basin of northwestern China

**Kaisheng Luo[1,2], Fulu Tao[1*]**

[1]Key Laboratory of Land Surface Pattern and Simulation, Institute of Geographical Sciences and Natural Resources

Research, Chinese Academy of Sciences, Beijing 100101, China

[2]University of Chinese Academy of Sciences, Beijing, 100101, China

*Corresponding Author: Fulu Tao (taofl@igsnrr.ac.cn)

**Abstract.** Climate variation will affect hydrological cycle, as well as the availability of water resources. In spite of large progresses have been made in the dynamics of hydrological cycle variables, the dynamics and drivers of blue water flow, green water flow and total flow (three flows), as well as the proportion of green water (GWC), in the past and future at

county scale, were scarcely investigated. In this study, taking the Heihe River basin in China as an example, we investigated the dynamics of green and blue water flows and their controlling factors during 1980-2009 using five statistical approaches and the Soil and Water Assessment Tool (SWAT). We found that there were large variations in the dynamics of green and blue water flows during 1980-2009 at the county scale. Three flows in all counties showed an increasing trend except

Jiayuguan and Jianta county. The GWC showed a downward trend in the Qilian, Suzhou, Shandan, Linze and Gaotai counties, but an upward trend in the Mingle, Sunan, Jinta, Jiayuguan, Ganzhou and Ejilaqi counties. In all the counties, the three flows and GWC had strong persistent trends in the future, which are mainly ascribed to rainfall variation. In the Qilian and Shandan counties, rainfall was the major controlling factor for the three flows and GWC. Rainfall controlled the green water and total flows in the Mingle, Linze and Gaotai counties; green water flow and GWC in the Suzhou county; green

water flow, total flow and GWC in the Jinta and Ejilaqi counties. Our results also showed that the "Heihe River basin allocation project" had significant influences on the abrupt changes of the flows above-mentioned. Our results illustrate the status of the water resources at county scale, providing a reference for water resources management of inland river basins.

## 1    Introduction

Much attention has been paid to the influence of climate variation on water resource all over the world (Jeuland and

Whittington, 2014; Zhang et al., 2015). In the context of global warming and increasing extreme weather frequency and intensify (Qiao et al., 2014; Zuo et al., 2015), studies have shown global climate change has led to reduction of water resource (Kundzewicz et al., 2008; Ravazzani et al., 2014), and exacerbated the shortage of water in the semi-arid regions. Water scarcity can endanger the food safety and sustainable development of economy, as well as the health of the ecosystem (Cheng et al., 2007; Piao et al., 2010).

Falkenmark (1995) first introduced the notions of blue water and green water and other researches afterwards progress it





(Falkenmark and Rockstrom, 2006; Faramarzi et al., 2009; Li et al., 2010; Schuol et al., 2008). In the view of stock, green water is reserved in soil from precipitation, and blue water comprises the water in the rivers, lakes, wetlands and shallow aquifers. From the perspective of flux, green water refers to actual evapotranspiration, and blue water includes the liquid water flows. Many scholars have undertaken related studies since the appearance of this notion (Faramarzi et al., 2009;

Jewitt et al., 2004; Liu et al., 2009b; Rost et al., 2008; Sulser et al., 2010; Xu and Zuo, 2014). Current reaches in the quantitatively assessing (Chen et al., 2015; Chukalla et al., 2015; Schuol et al., 2008) and theory frame develop, as well as related model designs (Faramarzi et al., 2009). But overall, previous studies focus mainly on the blue water and often overlooked the green water (Cao et al., 2014; Falkemnark, 1995). Furthermore, the previous studies mainly focused on the basin scale. County-level studies were rare. Meanwhile, little attention was paid to the controlling factors of blue water and

green water flows. In practice, there is a need for decision makers to understand the hydrological dynamics in each county. Therefore, the dynamics of blue water and green water flows on the county scale and their controlling factors deserve further investigation.

Dynamic analysis is extremely vital for the research in hydrology and meteorology, especially under the circumstance of global change (Karpouzos et al., 2010; Lopes and Machado, 2014). Generally, scenario analyses based on the hydrological

model and statistical test were two popular methods among many others. Scenario analysis method displayed many advantages in predicting the future variation because of using the hydrological models, Such as river discharge (Schneider et al., 2013), runoff (Piao et al., 2007). And the statistics were usually used to describe past dynamic (Feidas et al., 2004; Kundzewicz et al., 2008; Sen, 1968). Meanwhile, it's also quite vital to identify the driver of hydrological variation from the view of regional water resource management (Page et al., 2010). However, previous trend analysis has mainly focused on

runoff, which is a type of blue water flow (Blahusiakova and Matouskova, 2015; Hu et al., 2013; Liu et al., 2013), and potential evapotranspiration (Abtew et al., 2011; Li et al., 2015; Ye et al., 2013). It is very necessary to develop a hybrid dynamic detect approach that combines hydrological model and statistical methods based on their merits.

The Heihe River basin, the second largest inland river basin of China, was chosen as the experimental sit (Figure. 1). This basin with the fragile eco - environment has suffered serious water scarcity and endangered the safety of ecosystem and

human (Wang et al., 2010). Effective management of water resources is a crucial need in this basin. To achieve this aim, detailed and reliable information for each county of water flows is indispensable (Chen et al., 2006). Previous studies mainly focus on the dynamics of basin meteorological variables and runoff (Chen et al., 2010; Cheng et al., 2015; Li et al., 2011; Sang et al., 2014), and they rarely related to the green water and blue water flows. Therefore, we tried to analyze the dynamics and controlling factors of green and blue water on the county scale in this basin.

In this study, we assessed the variation of four core hydrological variables including blue and green water, the total flow,





and the proportion of the total accounted for by green water during 1980-2009. The aims are: 1) to analyze the dynamics of above-mentioned variables on county scale 2) to identify their controlling factors in each county.

## 2  Methods

### 2.1  Study area

The Heihe River Basin (HRB) lies within 38–42º N and 98–101º W in Northwest China and covers an area of $14.31 \times 10^5$ km$^2$ (Wu et al., 2013). Mean elevation is surpassed 1200 m and the range was 879-5573 m. The HRB comprises eleven counties which are the Qilian, Sunan, Gnazhou, Shandan, Suzhou, Mingle, Linze, Gaotai, Jinta, Jiayuguan, Ejilaqi. The upper HRB includes Qilian and Sunan county and the middle HRB includes Gnazhou, Shandan, Suzhou, Mingle, Linze, Gaotai county. The down HRB was consist of Jinta and Ejilaqi county. The basin mean rainfall is 193.4 mm a$^{-1}$ from 1980 to
2009. Mean annual rainfall declined from the upper basin to down basin (Yin et al., 2015).

### 2.2  Methods and data

The green water flow refers to actual evapotranspiration (AE), and the blue water flow comprises surface runoff, lateral flows, and groundwater recharge (Schuol et al., 2008). Green water coefficient (GWC) -the proportion of green water in total flows (blue and green flowers, TW) was used to measure the importance of blue and green flows (Liu et al., 2009a).

The amount of flows was simulated by the Soil and Water Assessment Tool (SWAT), which has successfully been applied to many countries and regions (Baker and Miller, 2013; Li et al., 2010; Schuol et al., 2008; Zende and Nagarajan, 2015). The Nash-Sutcliffe efficiency coefficient (NS), determination coefficient ($R^2$) and root mean square error (RMSE) -observations standard deviation ratio (RSR) was used to estimate the model simulation. Generally, model simulation is perfect when RSR range is between 0.00 and 0.50, and NS range is between 0.75 and 1.00. The simulation effect is good
when the range of RSR is from 0.50 to 0.60 and NS is from 0.65 to 0.75. The simulation effect is satisfactory when RSR belongs to 0.60-0.70 and NS belongs to 0.50-0.65 (Krause et al., 2005; Moriasi et al., 2007). One this base, we further assessed the accuracy of simulated green water by observed AE in the year of 2007.

In this study, land-use data come from the Chinese Academy of Sciences (CAS) Environmental Data Center. Monthly average discharge was obtained from the HRB Water Resources Administration. Daily climate data were from 16 weather
stations administrated by the China Bureau of Meteorology. The DEM of 30m resolution via the China Scientific Date Cloud. Soil data were from Harmonized World Soil Database version1. 1 via the FAO website.

### 2.3  Dynamic and controlling factor analysis

The Mann–Kendall (MK) trend test was selected to describe trends of flows, and the sequential Mann–Kendall (SMK) (Ahmad et al., 2015a; Ahmad et al., 2015b; da Silva et al., 2015) was chosen to identify abrupt variations. The MK test is
extensively applied to the trend analysis of long-time series of hydro-meteorological data because of its non-parametric



assumption, simpleness and stability (Ahmad et al., 2015b; Bae et al., 2008; Chen et al., 2014; da Silva et al., 2015). We followed other literature to calculate the MK and SMK (Ahmad et al., 2015b; da Silva et al., 2015). Theil–Sen method (TS) method was applied to character the change amplitude, which means the mean variation for each year. If the trend was displayed by linear, TS expresses the mean change rate per year. Determination coefficient of linear regression (DC) was

used to gauge the contributions of the independent variables to the dependent variables and find the controlling factor (Nagelkerke, 1991). In the unary linear regression, the independent variable can be identified as the controlling factor if DC > 0.50 (Bin et al., 2003; Nagelkerke, 1991). Figure. 2 listed the flowchart of our study.

## 3 Results

### 3.1 Model calibration and uncertainty analysis

The agreement between simulated and observed monthly discharge values in the periods of calibration and validation was excellent (Figure. 3). All NS and $R^2$ values were above 0.5, and absolute *SRS* values were below 25 (Table 1). These suggest model perform fairly well, in spite of small differences sometimes (Figure. 3). The results simulated by SWAT models can be applied to evaluate green water and blue water flows.

We used the observed annual AE in 2007 (Yang, 2009) to further estimate the green water flow due to the limitation of

observed data. The range of relative errors is from 3.51% to 5.78% (Table 2).

### 3.2 Dynamics of blue water flow

In the Qilian (Figure. 4a), Sunan (Figure. 4c), Ganzhou (Figure. 4e) and Mingle counties (Figure. 4g), the blue water flow increased from 1980 to 2009, although it was not significant. However, blue water flow at the Linze (Figure. 4k), Suzhou (Figure. 4m) and Gaotai counties (Figure. 4q) decreased significantly during this period ($p < 0.01$). Blue water flow

increased by 0.13, 0.68 and 0.55 per decade from 1980 to 2009 at the Linze, Suzhou and Gaotai counties, respectively. Meanwhile, the blue water flow increased significantly in Shandan county ($p < 0.05$), whereas decreased significantly at Jinta county ($p < 0.05$).

Based on the SMK test (Figure. 4), all the abrupt changes of blue water flow changed abruptly were mainly in the 1980s and especially 2000s except Gaotai county (Figure. 4r). The abrupt change over 2000s happened around 2000-2003,

indicating hydrological processes were subjected to large external interference around 2000-2003. For example, blue water flow at the Mingle (Figure. 4m) and Shandan counties (Figure. 4j) changed abruptly in 2001.

### 3.3 Dynamics of green water flow

In the upper basin including the Qilian county (Figure. 5a), the green water flow generally increased significantly from 1980 to 2009 ($p < 0.01$). Green water flow at Qilian county and Sunan county (Figure. 5c) increased by $1.32 \times 10^8$ m$^3$ and $0.13 \times 10^8$

m$^3$ per decade from 1980 to 2009, respectively. The blue water also generally increased in the middle-stream counties,





significantly at Ganzhou (Figure. 5e), Sunan and Shandan (Figure. 5j) counties (p < 0.01 at Ganzhou county and p < 0.05 at Shandan and Sunan counties). Blue water flow at Jiayuguan (Figure. 5o)) showed a weak decreasing trend from 1980 to 2009 (Figure. 5), versus an increasing trend at other ten counties. Blue water flow at the Ejilaqi (Figure. 5u) county in the downstream basin has a significant increasing trend from 1980 to 2009 (p < 0.01).

Green water flow at Qilian (Figure. 5b), Sunan (Figure. 5d), Ganzhou (Figure. 5f), Mingle (Figure. 5h), Gaotai (Figure. 5r)) and Ejilaqi (Figure. 5v) counties changed abruptly in 2001, 2003, 2003, 2001, 2001 and 1985, respectively (Figure. 5). It changed abruptly in 1996, 2001 and 2005 at Jita county (Figure. 5t), and in 1982 and 2005 at Jiayuguan county. It changed abruptly in 1983, 1986 and 1993 at Suzhou county (Figure. 5n). However, there were always abrupt changes followed 2000, around 2000-2003. Meanwhile, in the study period blue water flow has different trends. For example, at the Linze county (Figure. 5l), blue water flow increased from 1980 to 1982, decreased from 1982 to 1985, and then increased from 1985 to 1995, decreased from 1995 to 2001, and decreased from 2001 to 2004, and finally increased from 2004 to 2009.

The GWC generally decreased from 1980 to 2009 at Qilian, Suzhou, Shandan, Linze and Gaotai counties, but insignificantly. Meanwhile, the GWC increased at Mingle, Sunan, Jinta, Jiayuguan and Ganzhou counties (Table 3). The dynamics of GWC generally are same as those of green water flow, although not completely consistent. The reason for this is that the GWC depend on green water flow and total flows, not only green water flow.

### 3.4 Dynamics of total flow

The trends in the blue water, green water, and total flows showed a considerable spatial variation among the counties. The MK test showed that total flows generally increased in nine counties, including the Suzhou, Shandan, Qilian, Mingle, Linze, Sunan, Gaotai and Ejilaqi (Table 4). The increase was significant in Mingle, Gaotai and Ganzhou from 1980 to 2009 (p < 0.05; Table 4). The largest increase was in the Ejilaqi county, where total flow increased by $6.17 \times 10^8$ m$^3$ from 1980 to 2009, equivalent to a rate of increase of $2.08 \times 10^8$ m$^3$ per decade. This was caused by an increase in green water flow (Figure. 5). However, the total flows at Jinta, and Jiayuguan decreased insignificantly in the past three decades (Table 4). This is mainly attributed to decrease in blue water flow (Figure. 4). Blue water flow decreased at Jinta and Jiayuguan by $0.07 \times 10^8$ m$^3$ per decade and $0.33 \times 10^8$ m$^3$ per decade, respectively (Figure. 4).

### 3.5 Predicted trends in blue water, green water and total flows in the future

In the upstream including the Qilian county (Figure. 6a), the hydrological variables have insistent trends in the future, with H > 0.50 (Figure. 6). The strength of the persistence of trends in descending order is: green water flow > total flow > GWC > blue water flow. All the variables have strong persistent trends, will keep increasing in the upstream basin.

In midstream basin, including Sunan (Figure. 6b), Ganzhou (Figure. 6c), Mingle (Figure. 6d), Shandan (Figure. 6e), Linze (Figure. 6f), Suzhou (Figure. 6g), Jiayuguan (Figure. 6h) and Gaotai (Figure. 6l) counties, all hydrological variables have




persistent future trends, with H > 0.5 (Figure. 6). This indicates that the blue water flow, green water flow, total flow and GWC will keep increasing. At Jiayuguan, Sunan and Jinta county, the Hurst index of blue water flown were close to the value of 1.00, suggesting that persistent trends are obvious, and the blue water flow in both counties should continue to decrease in future. However, the hydrological variables at Shandan county do not have an obvious persistent trend with the

5        Hurst index of green water flow and total flow closing to 0.50, which represents quite independent trends.

In the downstream basin, including Jinta and Ejilaqi counties, the trend for blue water flow has an obviously persistent change in the future, with H=0. 99 at Jinta county (Figure. 6j) and H=0. 98 at Ejilaqi county (Figure. 6k). This means that the blue water flow may quickly decrease in the future. The Hurst indexes for the Green water flow, total flow and GWC also have persistent future trends, with H > 0.5. This implies that those hydrological variables will increase in the downstream

basin. The most persistent change trends display in the downstream basin.

All of Hurst index values of blue water flow were surpassed 0.5, which indicates that the trends in the past were likely to continue (Figure. 6) The counties including Qilian, Sunan, Ganzhou, Shandan, Mingle, Linze, Suzhou and Ejilaqi have persistent trends and will keep increasing in the future, whereas blue water flow at Jiayuguan and Jinta counties may decrease in the future inconsistent with the past trend.

For eleven counties in the HRB, the blue water flow, green water flow, total flows, and GWC generally have continuous future trends as in the past, with H > 0.5 (Figure. 6), especially in the upper and middle HRB.

### 3.6   Controlling factors for dynamics of blue water, green water and total flows

For the blue water flow, the trends and influence factors differ for different counties of the river basin among eleven counties. One reason for this is that rainfall generally increased and temperature significantly increased during this period (Table 5 and

Figure. A1). We detected that rainfall was the controlling factor at Qilian, Sunan and Shandan county (Table 6, DC=0. 90 at Qilian and DC=0. 81 at Shandan), where their general trends and the fluctuations were closely related to their variations of rainfall (Figure. A1). Rainfall contributed more to blue water flow than the temperature in Mingle, Linze, Jiayuguan and Ganzhou. By contrast, the contribution of temperature was more than that of rainfall in Suzhou, Jinta, Gaotai, Ejinaqi. However, neither the rainfall nor the temperature was the main controlling factor in these nine counties (Table 5). Human

activities may be the mainly driver for land-use change.

For the green water, obviously, our analysis showed that the controlling factor was the rainfall with all DC>0.84 during the period of 1980-2009 (Table 6). This controlling action of rainfall is extremely obvious in Linze, Jinta, Gaotai and Ejilaqi counties with DC>0.90.

### 4   Discussion

The blue water for counties in the HRB generally increased from 1980 to 2009, although the blue water flow at Jiyuguan




county and Jinta county decreased insignificantly (Figure. 4). Qilian in the upstream basin was the runoff-product area for the whole basin, where the rainfall was generally high. An increase in rainfall led to a fast increase in runoff (Li et al., 2011). The barrier effect of the Qilian Mountains in the upstream basin had caused different regional hydrological cycles and different rainfall sources (Jia et al., 2008). Rainfall in upstream was influenced by different atmospheric circulations that resulted from local topography, i.e. the West Pacific Subtropical High and by the Indian Ocean Southwest Monsoon (Jia et al., 2008). The West Pacific Subtropical High and the Indian Ocean Southwest Monsoon began to change in the 1980s, leading to an abrupt change in rainfall in 1980s (Jia et al., 2008; Lan et al., 2001). In Shandan county, blue water flow increased significantly from 1980 to 2009. But they also fluctuated in different periods. The channeling effect at this county resulted in different regional hydrological cycles and rainfall sources (Lan et al., 2001). Hence, at Shandan, rainfall was influenced by different atmospheric circulations and topography (Lan et al., 2001). Rainfall there was mainly influenced by the Western Pacific Subtropical High and by the Eurasian Middle–High Latitude Circulation, which led to fluctuations of rainfall at Shandan county.

The land use change maybe the driver of blue water in Mingle, Linze, Jiayuguan, Ganzhou, Suzhou, Jinta, Sunan, Gaotai, Ejinaqi counties. Surface roughness, albedo and other properties that affected the exchange of water and energy between the earth surface and atmosphere could be altered by conversions of land use, resulting in variability of surface energy and net radiation (Kueppers and Snyder, 2012), and further influencing hydrological processes. Land use change, especially the expansion of agricultural land, had influenced on hydrological processes and water balance of Heihe River basin (Deng et al., 2015; Fu et al., 2014; Liu et al., 2010; Nian et al., 2014). The expansion of farmland had a significant influence on the shallow aquifer water system through intensive irrigation. With the expansion of farmland, ground water recharge increased due to enhanced infiltration in these counties (Wang et al., 2014). Deforestation also reduced soil water retention capacity and raise surface runoff (Zhou et al., 2002).

Climate will have a large influence on future blue water flow. Climate models predicted a weak increase in annual rainfall and significant increase in temperature (Dahe et al., 2002). Climate change is likely to cause annual runoff variation (Zhi et al., 2009) and increase surface runoff by more than 10% during 2010-2050 (Dahe et al., 2002). An increase in rainfall will rarely produce large amounts of runoff because of the flatter topography combined with the high potential evapotranspiration and low soil moisture content.

The green water depended mainly on water availability and air temperature in the northwest of China (Han et al., 2012; Yang and Yang, 2012). Since 1850, the temperature on the Earth's surface has been increasing continuously; climate warming is an obvious phenomenon (Stocker, 2014). In general, because of climate warming, the air near the earth's surface would become drier, which further leads to an increase in AE (green water) (Cong et al., 2009). Nevertheless, variation of





AE is of significant importance for hydrologic water cycle (Zhang et al., 2012). In the arid or semi-arid, the green water was limited by water availability (Sun and Wu, 2001). Green water was not strongly driven by temperature, but the rainfall, although there are significant changes in temperature in the study period. An increase in rainfall will increase surface runoff and soil moisture, leading to the increase in blue water flow (AE). In the Ganzhou county, both rainfall and temperature have

a positive influence, leading to increase in green water. The contribution of rainfall was much larger than that of temperature. However, in the Ganzhou county, the contributions of rainfall and temperature were low, and both were not the main driver. This suggests that human activities should be the controlling factor of green water in the Ganzhou county, which offset the positive influence induced by the increase of rainfall and temperature. Ganzhou county was the commodity grain base for the whole Gansu province and agriculture land depended on irrigation quite wild (Chen et al., 2006). For Ganzhou county, the

rapid expansion of irrigation in the past three decades has significantly altered the energy budget at the land surface (Wisser *et al.*, 2009). AE will change when irrigation related change in soil moisture, deep filtration and underground recharge (Wang et al., 2014). Our results indicate that green water flow will keep trend in the past, all the counties will increase expect Jiayuguan county where the future trend will decrease (Figure. 5o and 6h).

When green and blue water flows are summed, the trend in the total flow appears to reflect a combined influence of climate and human activities in the eleven counties in the HRB from 1980 to 2009 (Table 4). Because the green water flow

was much more than blue water flow in the HRB, the general trends in total flow were similar to green water flow. It increased in Qilian, Ganzhou, Sunan, Suzhou, Linze, Gaotai, Shandan, Mingle and Ejilaqi counties, whereas decreased in Jinta and Jiayuguan counties. Meanwhile, our analysis showed the controlling factor of total flow at each county was different from that for green water flow: the total flow in the most counties was mainly influenced by rainfall (DC＞0. 5;

Table 6), however, at Jinta and Jiayuguan county, the human activities were the major controlling factor. With increases in rainfall and temperature, the increase in blue water and green water flows lead to the increase in the total flow at Qilian, Ganzhou, Sunan, Suzhou, Linze, Gaotai, Shandan, Mingle and Ejilaqi counties. But the total flow decreased due to decrease in blue water flow and green water flow, which was induced by increases in rainfall and temperature at Jiayuguan county. Our results also showed that the decrease in blue water flow and green water flow at Jinta county would increase blue water

more than green water, leading to a decrease in total flow. Hence, it would be logical to predict that there is a persistent trend for total flow in the eleven counties, which is consistent with the trend in green water driven by rainfall expect Jinta and Jiayuguan counties (H＞0. 5, Figure. 6).

There are variations in the dynamic and controlling factor of the GWC in different counties in the HRB (Table 5 and 6). For all of the counties, the temperature was not the controlling factor and the contribution of rainfall to the GWC is relatively

higher. The GWC increase induced by green water rapidly rise with the increase of rainfall and temperature in Linze, Suzhou,





Shandan, Ejilaqi county. The increase of rainfall and temperature in the Jiayuguan and Jinta county would increase runoff (blue water) more than evapotranspiration (green water), leading to a lower GWC. For Qilian, Suzhou, Shandan, Jinta, Jianyuan and Ejilaqi counties, the rainfall determined the trends and variability of the GWC. All the Hurst index values surpassed 0.50, indicating the GWC depend on the past trend and will keep the trend in the future.

Hence, the rainfall exerted the most influence on three flows in the HRB. The different dynamic of three flows formed caused by different generation mechanisms of rainfall on the county scale. Therefore, the abrupt changes of temperature and rainfall could have influenced both the blue water and total flows. Moreover, the flows are not only influenced by atmospheric circulation, but also by local human activities, such as the blue water flow of Mingle, Linze, Jiayuguan and Ganzhou counties. The quantitative impacts of human activities in each county will also need further research. The similar

abrupt changes in the water flows, rainfall, and temperature reflect the close relationship between these three parameters in the eleven counties.

At the county level, different rainfall patterns also can lead to different hydrological processes. For example, the rainfall decreased in Shandan county significantly ($P<0.05$), but increased in Mingle county. However, the difference caused the blue water flow to increase in both counties (significantly for Shandan county, and insignificantly for Mingle county). In the

future, the three flows will have an increasing trend due to the projected increase of rainfall.

The dynamic of the climate can largely explain the variations and abrupt changes of three flows in different periods (Table A1and Figure. A1). However, there are some abrupt changes in the 1990s and 2000s, which were close to 1992, 1997 and 2000 when rainfall and temperature had no abrupt changes in the corresponding periods. These abrupt changes should be attributed to the China's water transport project "Heihe River basin allocation policy" initiated in 1992,1997 and 2000 to

ensure the enough water availability and in case of drying up of Dongjuyanhai Lake. The water transport project makes an influence on green water flow and blue flown mainly though changing the underground water level and surface water. "Heihe River basin allocation policy in 2000 had decreased ground water level by 1.3-2.7m in the middle HRB (Haiyang et al., 2007) and raised it by 0.4-0.6m in the down HRB (Zhi et al., 2007). Yinchun et al (2014) indicated the water transport project deceased the number of water withdrawal gates by 26%, reduced the blue water withdrawal by $1.05\times10^9$ m$^3$ and

increased the underground water exploitation by $1.64\times10^9$ m$^3$ in the whole Heihe River basin (Yinchun et al., 2014). In the middle-stream, "Heihe River basin allocation project" in 2000 induced the decrease the underground water level in the irrigation region caused by the reduction of exchange of surface and underground water, and expansion of the underground water exploitation (Zhi et al., 2008).

This study has some limitations. First, more data need to be obtained and complemented, such as the hydrological data in

the lower reaches of the basin and the meterological data for the whole basin. Second, the impacts of human activities need





to be quantified. In addition, we also need to note that a nuance existed between the methods used in this paper and others for capturing the abrupt changes, and that there exists an alternative method, that is, calibrated SWAT model is driven by future climate scenarios to predict the future dynamics of hydrological elements. It would be interesting to compare the differences between the two methods.

## 5    Conclusions

We investigated the dynamic of water flows and GWC in the HRB by five statistical tests with SWAT model on the county scale. We further detected the controlling factor for these trends for each county.

(1) For the Qilin, Suzhou, Shandan, Mingle, Linze, Gaotai, Ganzhou, Sunan and Ejilaqi counties, the blue water, green and total flows have increased from 1980 to 2009 and these trends were significant in above-mentioned counties, especially

for the blue water flow. For the Jiayuguan county, the blue water, green and total flows have decreased from 1980 to 2009. For the Jinta county, green water and total water flows have decreased from 1980 to 2009, whereas the blue water had a weak increase trend. The GWC presented some decrease in Qilian, Suzhou, Shandan, Linze, Gaotai county, and an increasing trend was found in the Mingle, Sunan, Jinta, Jiayuguan, Ganzhou and Ejilaqi counties. In all the counties in the HRB, three flows and GWC have persistent future trends and the strength generally were quite large, especially in

the Sunan and Ejilaqi counties.

(2) These trends are ascribed mainly to rainfall due to the limitation of water availability in the semiarid HRB. At Qilian and Shandan counties, rainfall was the controlling factor of the blue water flow, green water flow, total flow and GWC. At Mingle, Linze and Gaotai counties, the Green water and total flows were determined by rainfall. At Suzhou county, the trends of green water flow and GWC mainly were controlled by rainfall. At the Jinta and Ejilaqi county, the

variation of green water flow, total flow and GWC mainly were driven by rainfall. Meanwhile, the human activities, especially land use change, appear to be the dominating cause of these trends of tree flow and GWC in Shandan county. However, the quantitative influences of anthropogenic activities will need further study.

(3) The abrupt changes in the three water flows and in GWC were mainly affected by the China's water transport project "Heihe River basin allocation project" respectively initiated in 1992,1997 and 2000, although rainfall and temperature

make some influences, especially in the 1980s. "Heihe River basin allocation project" has an important influence on the hydrological processes and three water flows in the HRB.

**(4)** This study showed the past and future dynamic of hydrological variable for eleven counties in the HRB and their controlling factors, which can improve the understanding of human and in favor of water resources management for the decision makers. More attention should be paid into the counties where the water availability and GWC declined. Thus,

this study provides the reference for further studies on the county scale in similar regions and basin. Variations in the





dynamic and controlling factors in different counties suggested decision maker should adopt different policies in different counties in the basin.

**Acknowledgements**

Funding for this study was provided by the project of China Natural Science No.91325302. The data for this paper are available at the websites, official publication and publication referenced in the text.

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

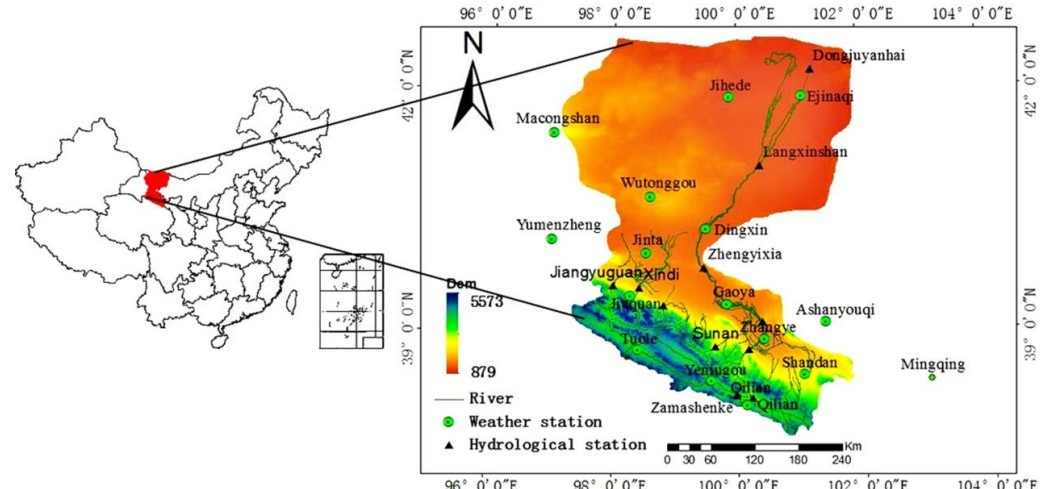



**Figure 1:** A map depicting the location of the Heihe River Basin in China (left plate) and the expanded study area (right plate) depicting

the elevation, river and data stations used in the study.

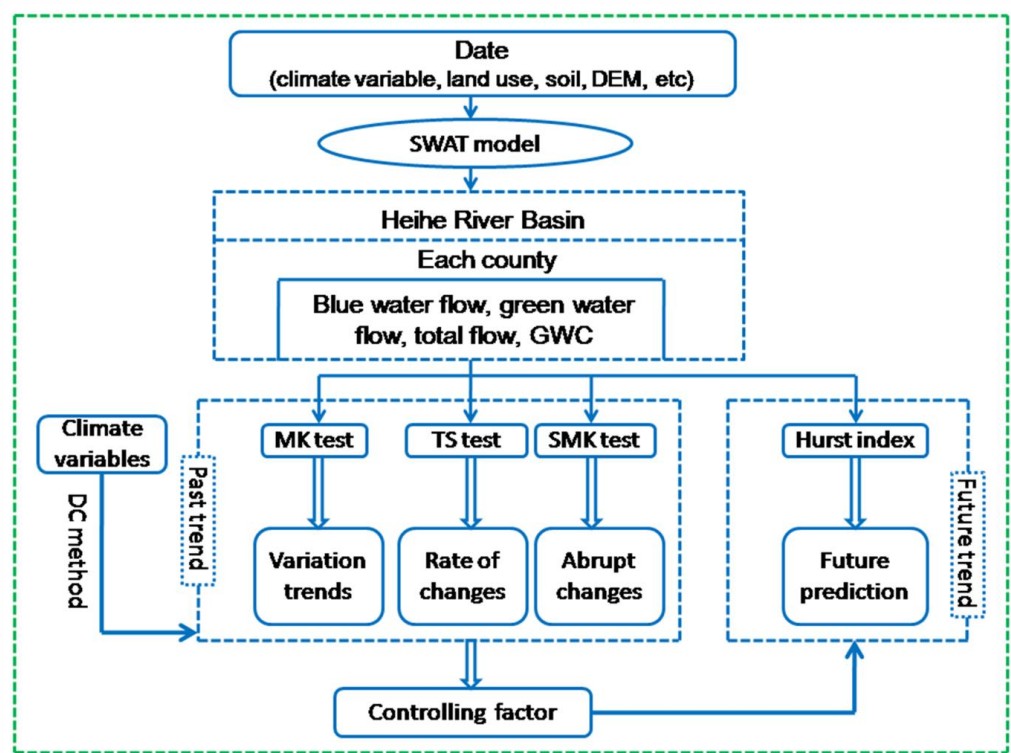

**Figure 2:** Flowchart of this study. Mann–Kendall test is represented by MK; SMK is Sequential Mann–Kendall test represented by SMK;

5 TS represents Theil–Sen method. DEM is a digital elevation model; GWC is the proportion of green water in the total flows (blue and

green flows); SWAT is Soil and Water Assessment Tool.





**Figure 3:** Monthly simulated and observed discharge in the Heihe River Basin during 1981–2010; Model validation period and calibration period of Zamashene (a), Yinluoxia (b), Zhengyixia (c) was respectively 1981–1997 and 1998–2012); Model validation period and calibration period of Langxinshan (d) and Dongjuyanhai (e) was respectively 2006-2010 and 2011-2012.





**Figure 4:** Dynamics of blue water flow at eleven counties of the Heihe River basin, including Qilian (a and b), Sunan (c and d), Ganzhou (e and f), Mingle (g and h), Shandan (i and j), Linze (k and l), Suzhou (m and n), Jiayuguan (o and p), Gaotai (q and r), Jinta (s and t) and Ejilaqi (u and v). Note: TS stands for the Theil-Sen test; *** indicates the significant level is at $p < 0.01$; + indicates the significant level is at $p < 0.05$; NS indicates the trend is not significant; $Z > 0$ indicates a downward trend and $Z < 0$ indicates an upward trend; Q stands for the change rate; $u(t) > u(t)'$ denotes increase, while $u(t) < u(t)'$ denotes decrease.






**Figure 5:** Dynamics of green water flow for eleven counties of the Heihe River basin, including Qilian (a and b), Sunan (c and d),

Ganzhou (e and f), Mingle (g and h), Shandan (i and j), Linze (k and l), Suzhou (m and n), Jiayuguan (o and p), Gaotai (q and r), Jinta (s

and t) and Ejilaqi (u and v). Note: TS stands for the Theil-Sen test; **indicates the significant level is at p < 0. 01; * indicates the

5    significant level is at p < 0. 05; NS indicates the trend is not significant; Z > 0 indicates a downward trend and Z < 0 indicates an upward

trend; Q stands for the change rate; u (t)  > u (t)' denotes increase, while u (t)  < u (t)' denotes decrease.





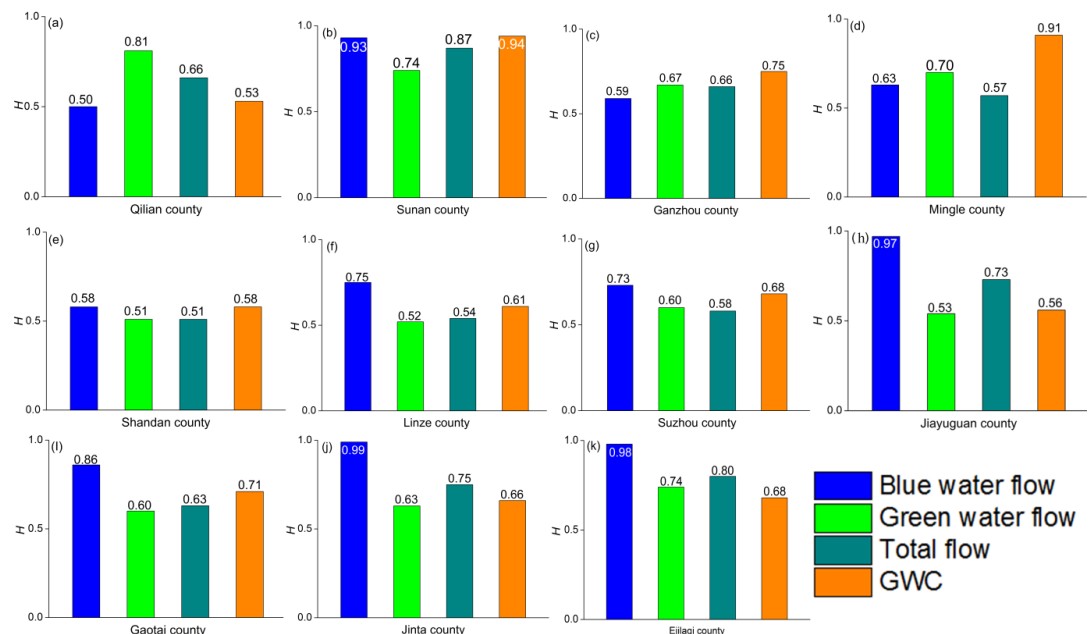

**Figure 6:** The Hurst index (H) for trends in three flows (blue water, green water and total flows) and the GWC (the proportion of the total flows accounted for by the green water flow) for eleven counties of the Heihe River basin. Values of H > 0. 5 indicate the trend from 1980 to 2009 will continue in the future; while H < 0. 5 indicate the trend will change direction.

5                              **Table 1.** SWAT model performance during the calibration and validation period

| Station | Calibration | | | Validation | | |
| --- | --- | --- | --- | --- | --- | --- |
| | $NS$ | $R^2$ | $RSR$ | $NS$ | $R^2$ | $RSR$ |
| Zamashene | 0.76 | 0.92 | 0.08 | 0.74 | 0.90 | 0.09 |
| Yingluoxia | 0.71 | 0.85 | 0.10 | 0.73 | 0.80 | 0.12 |
| Zhenyixia | 0.68 | 0.61 | 0.13 | 0.63 | 0.55 | 0.16 |
| Langxinshan | 0.52 | 0.62 | 0.19 | 0.56 | 0.57 | 0.18 |
| Dongjuyanhai | 0.52 | 0.53 | 0.17 | 0.51 | 0.52 | 0.22 |

**Table 2.** Comparison between the simulated and observed annual green water flow in 2007 (Units: mm a$^{-1}$)

| Stations | Observed ET | Simulated ET | Relative error |
| --- | --- | --- | --- |
| Jinta | 114 | 110 | 3.51% |
| Gaotai | 101 | 97 | 3.96% |
| Zhangye | 188 | 181 | 3.72% |




| Linze | 173 | 183 | 5.78% |
| Shandan | 136 | 141 | 3.68% |

**Table 3.** The results of the Mann–Kendall test (MK), sequential Mann–Kendall test (SMK), and Theil–Sen estimator test (TS) for eleven counties of GWC in the Heihe River basin.

| Counties | MK | SMK | TS | | MK | SMK | TS |
|---|---|---|---|---|---|---|---|
| Qilian | NS↓ | 1984↑2009↓ | -0.010 | Jinta | NS↑ | 1985↓2003↑ | 0.100 |
| Suzhou | NS↓ | 1984↓ 2006↓ | -0.020 | Jiayuguan | NS↑ | 1983↓2006↑ | 0.010 |
| Shandan | NS↓ | 2001↓ | -0.050 | Gaotai | NS↓ | 1982↓ | -0.200 |
| Mingle | NS↑ | 2006↑ | 0.037 | Ganzhou | NS↑ | 2004↑ | 0.180 |
| Linze | NS↓ | 2006↓ | -0.09 | Ejilaqi | NS↑ | 1996↓2004↑ | 0.170 |
| Sunan | S(*)↑ | 1984↑ 2009↓ | 0.024 | | | | |

Note: TS stands for the Theil-Sen test and the value of presents the change rate; ↑ indicates an increasing trend, whereas ↓ indicates a

5  decreasing trend; * indicates the significant level at $p < 0.05$; NS represents the test is not significant.

**Table 4.** Dynamics of total water flow in the eleven counties in Heihe River basin.

| Counties | MK | SMK | TS | Counties | MK | SMK | TS |
|---|---|---|---|---|---|---|---|
| Suzhou | NS↑ | 1998↑ | 0.037 | Jinta | NS↓ | 2004↓ | -0.034 |
| Shandan | NS↑ | 2001↑ | 0.048 | Jiayuguan | NS↓ | 2000↓ | -0.004 |
| Qilian | NS↑ | 2007↑ | 0.175 | Gaota | S(*)↑ | 1986↑ | 0.0700 |
| Mingle | S(*) | 2004↑ | 0.031 | Ganzhou | S(*)↑ | 2003↑ | 0.020 |
| Linze | NS↑ | 2004↑ | 0.027 | Ejilaqi | S(*)↑ | 1984↑ | 0.250 |
| Sunan | NS↑ | 1984↓2006↑ | 0.020 | | | | |

Note: MK represents Mann–Kendall test; SMK represents sequential Mann–Kendall test; TS stands for the Theil-Sen test and its values

represent the change rate; ↑ declares an increase trend and ↓declare an decrease trend; *signals a significance at $p < 0.05$; NS represents

the test is not significant.

10  **Table 5.** Coefficients of determination (DC) of mean temperature for three flows and GWC (proportion of green water flow in total flows) in eleven counties of Heihe River basin during 1980-2009.

| Variables | Qilian | Suzhou | Shandan | Mingle | Linze | Jinta | Jiayuguan | Gaotai | Ganzhou | Ejilaqi | Sunan |
|---|---|---|---|---|---|---|---|---|---|---|---|
| Blue water flow | 0.01 | 0.15 | 0.01 | 0.03 | 0.05 | 0.06 | 0.02 | 0.15 | 0.03 | 0.03 | 0.01 |





| Green water flow | 0.31 | 0.04 | 0.02 | 0.02 | 0.01 | 0.02 | 0.04 | 0.03 | 0.06 | 0.02 | 0.11 |
| Total water flow | 0.09 | 0.08 | 0.02 | 0.04 | 0.01 | 0.01 | 0.03 | 0.07 | 0.06 | 0.02 | 0.03 |
| GWC | 0.01 | 0.01 | 0.02 | 0.03 | 0.03 | 0.03 | 0.03 | 0.02 | 0.03 | 0.03 | 0.04 |

**Table 6.** Coefficients of determination (DC) of rainfall for blue water flow, green water flow, total water flow in eleven counties in the Heihe River basin in the past three decades.

| Variables | Qilian | Suzhou | Shandan | Mingle | Linze | Jinta | Jiayuguan | Gaotai | Ganzhou | Ejilaqi | Sunan |
|---|---|---|---|---|---|---|---|---|---|---|---|
| Blue water flow | 0.90 | 0.03 | 0.81 | 0.32 | 0.068 | 0.03 | 0.03 | 0.038 | 0.21 | 0.02 | 0.70 |
| Green water flow | 0.56 | 0.88 | 0.84 | 0.85 | 0.96 | 0.90 | 0.86 | 0.94 | 0.25 | 0.97 | 0.80 |
| Total water flow | 0.90 | 0.31 | 0.88 | 0.81 | 0.90 | 0.78 | 0.6 | 0.62 | 0.32 | 0.91 | 0.85 |
| GWC | 0.77 | 0.81 | 0.60 | 0.22 | 0.44 | 0.79 | 0.82 | 0.28 | 0.03 | 0.90 | 0.39 |