# Peer review of "Dynamics of green and blue water flows and their controlling factors in Heihe River basin of northwestern China"

_Hydrology and Earth System Sciences, 2016_

## Referee Comment (RC1) · Anonymous Referee #1 · 27 Aug 2016

The manuscript tackles the issues of assessing (a) blue and green water fluxes as well as (b) the key factors controlling these. It is framed as a textbook application of very well known techniques to specific settings, as rendered by diverse counties in a part of China. Basically, the authors ground their analyses on the use of the Soil and Water Assessment Tool (SWAT) and employ diverse metrics (which are not 5 statistical approaches, as claimed by the authors) to assess the goodness of their results. Frankly, the level of scientific originality of the study, in terms of theoretical developments and conceptual advancement, is rather limited. Additionally, the relevance of the diverse quantities which are considered as potential controlling factors is tested with methods which do not assess the contribution of each of these factors to the variability of the

target state variable. As such, it provides at best an incomplete picture and does allow to draw general conclusions nor to obtain a robust and quantifiable uncertainty assessment. It is also not clear how the authors include quantitatively issues such as measurement uncertainty in their analyses.

Given the above issues, and considering the significantly application-oriented nature of the work, I would recommend it to be released from HESS. Implementing a robust uncertainty analysis as well as innovations in the methods employed would require a set of revisions going beyond the scope of major revisions. I think the authors would be best served if they submit their work to a more application-targeted Journal. I am afraid I cannot be more positive at this time.

---

## Referee Comment (RC2) · Anonymous Referee #2 · 29 Aug 2016

The focus of the proposed study is on green and blue water flow dynamics and their controlling factors in the Heihe basin in China. The scope of the study is without doubt of broad interest for the audience of HESS. However, I regret to say that neither the proposed methodology nor its presentation meet the standards required for a publication in HESS. The author present mainly an application of SWAT in combination with different standard statistical methods to assess trends in simulated green and blue water components. The presentation of the model results is very brief (0.75 pages), the author mainly declare that their model is suited for the objective of their study and start straight away discussing the trends in simulated flows.

Contrary to the author's statement, I found their results are not so convincing. For instance the authors evaluate their model on a monthly basis, regardless of the fact that river discharge reveals a strong seasonal pattern. In such a case, and as discussed by Schaefli and Gupta (2007), the null model is not the overall mean of the discharge, but the mean annual cycle of daily discharges. A proper evaluation implied either to benchmark the model against predicted deviations from the annual cycle, or to work at the daily scale. Secondly, the authors compare simulated and annual ET totals to underpin that their model is well suited to discriminate green and blue water flows. Firstly, I wonder how the authors estimated ET annual totals, unfortunately the manuscript does not provide any information on the data sources which are used to drive and test the model. If their ET estimation is based on the long term water balance (P-Q), this cannot be regarded as independent assessment. Secondly, one cannot conclude to have a model which reproduces green water flow dynamics well, without comparing the model against dynamic data.

In conclusion, I cannot recommend further treatment of this submission in HESS. The authors might consider either to largely enhance the scientific depth of their study or to submit their work to a more applied journal.

References: Schaefli, B., and Gupta, H. V.: Do nash values have value?, Hydrological Processes, 21, 2075-2080, 10.1002/hyp.6825, 2007.

---

## Author Comment (AC1) · 12 Sep 2016

Dear Sir: We are very grateful for your insightful comments and suggestions, which have improved our manuscript. We have revised the manuscript based on your comments. We hope that it would be accepted for publication in HESS. Our responses are in blue. Thank you once again. Yours truly, Fulu Tao

1 Comment: Basically, the authors ground their analyses on the use of the Soil and Water Assessment Tool (SWAT) and employ diverse metrics (which are not 5 statistical approaches, as claimed by the authors) to assess the goodness of their results. Response: Yes, we applied 5 statistical approaches, including the Mann–Kendall (MK) trend test, the sequential Mann–Kendall (SMK), Theil–Sen method (TS), Determination

coefficient of linear regression (DC) and Hurst index (H) to validated the use of SWAT model for the basin. 2 Comment: The level of scientific originality of the study, in terms of theoretical developments and conceptual advancement, is rather limited. Response: Traditional studies on the water resources pay little attention to the ecology water use, so Falkenmark (1995) first introduced the notions of blue water and green water and this scope became the research hot spot. Many scholars have undertaken related studies since the appearance of this notion and this scope has made great progress. However, county-level studies of green and blue water flow were rare. County-level blue and green analysis is quite important and informative for water managers to formulate specific and suitable strategies. The regional/basin studies can give us a big picture, and develop adaptive strategies for addressing the possible risks does need the local studies because all these strategies/schemes need to be implemented at level of county. In practice, there is an urgent need for decision makers to understand the green and blue water flow in each county. However, little previous researches focused on this scale and there is lack of a method and framework for the assessment of green and blue water flow at county-level. In this study, we specially focused on the county scale and develop a framework for the assessment of water resources including blue and green water and the total flow, combing SWAT hydrological model and statistical methods. Based on this framework, we mainly analyzed the dynamics of above-mentioned water variables for eleven counties in the Heihe River Basin of China during 1980-2009 and further identified their controlling factors in each county. There is little knowledge of county-level green and blue water flow, especially the controlling factors. Our study found that the dynamic and controlling factors vary considerably over counties, which are different from the region-level and basin-level findings. Meanwhile, this study provides the reference for further studies on the county scale in other regions and basins. Therefore, this paper is supposed to contribute greatly to theoretical developments and conceptual advancement. 3 Comment: Additionally, the relevance of the diverse quantities which are considered as potential controlling factors is tested with methods which do not assess the contribution of each of these

factors to the variability of the target state variable. Response: A large number of publications in the literature deal with the influence factors of water resources, e.g., green flow and blue flow on global or regional scales. However, there is currently only limited knowledge which factor control the green and blue water flow on county scale. Specially, the county-level assessment of blue and green water flows was lack of a standardized evaluation method and cannot obtain much reference from previous literature. But there are many questions needed for satisfactory answers. Whether the dynamics of green and blue water are inconsistent with the basin and region? What spatial variations are there in the dynamics of county-level green and blue water flow? What future trends in these dynamics? Which is the controlling factor for each county within basin? What is the pattern of controlling factors within a basin? In this paper, we have tried our best to overcome these difficulties and answer the above mentioned concerns in study area yet. Here, our work is supposed to much contribute to study on the assessment method and contents of county-level water resources. Therefore, our study can be perceived to be fully innovational and insightful. Most of time, it is enough to know the controlling factor of county-level water resources for policy makers and water resource managers. It would be interesting to quantify the contribution of the influence factors of county-level green and blue water flows in further study in future, which calls for more experiments and new method to make it. However, it is beyond the topic in this paper. 4 Comment: As such, it provides at best an incomplete picture and does allow to draw general conclusions nor to obtain a robust and quantifiable uncertainty assessment. It is also not clear how the authors include quantitatively issues such as measurement uncertainty in their analyses. Response: I would like to say that it is impossible to build a complete picture though a study only, because any progress in science is attributed to many experiments from so many researches over long time. The aim of our work is to address some problems and make some progresses in this scientific scope and hope this study served as a wake-up call to the scientific community and more scholars alike to focus on the study of county-level water resources. Our conclusions obtained from experiment are new and interesting,

rather than general conclusions. Those conclusions were displayed in the Conclusions Department of manuscript. Such as the abrupt changes in the three water flows were mainly affected by the China's water transport project "Heihe River basin allocation project" respectively initiated in 1992, 1997 and 2000. For instance, at Qilian and Shandan counties, rainfall was the controlling factor of the blue water flow, green water flow, total flow. Please see the Conclusions Department in manuscript. As for the measurement uncertainty, in fact, we do many works during experiments which include yearly and monthly calibration and validation of hydrological model and comparing the simulated evapotranspiration (green water) with measured evapotranspiration by remote sensing technology. Given the length limit of manuscript, the state is not enough detail in 3.1 department of manuscript. In this paper, the Nash-Sutcliffe efficiency coefficient (NS), coefficient, percent bias (PBIAS) and RMSE-observation standard deviation ratio (RSR) (Awan and Ismaeel, 2014; Krause et al., 2005; Moriasi et al., 2007; Troin and Caya, 2014) were used to assess the reliability and accuracy of the model simulation. Meanwhile, as discussed by Moriasi and Arnold (2007), on the monthly scale, a model simulation is rated as good if $0.65 < NS < 0.75$, $0.50 < RSR < 0.60$ and $\pm10\% < PBIAS < \pm15\%$. A model simulation is judged as satisfactory if $0.50 < NS < 0.65$, $0.60 < RSR < 0.70$ and $\pm15\% < PBIAS < \pm25\%$ (Moriasi et al., 2007). Base on this, we did not present the results of uncertainty analysis on year scale. As a matter of fact, the hydrological performance is better and the uncertainty is smaller on the year scale than that on the month scale. In the revised version, we added the content in yearly uncertainty estimate and in more details and stated uncertainty analyses in 3.1 section. 5 Comment: Given the above issues, and considering the significantly application-oriented nature of the work, I would recommend it to be released from HESS. Response: Response: In this paper, the aim is to propose a new method and develop a new framework for the assessment of green and blue water at the county level, combing SWAT hydrological model and statistical methods. The application in Heihe River Basin of China is to carry out and verify the new method and framework. Our work also conforms to the aim and scope of HESS

(http://www.hydrology-and-earth-system-sciences.net/about/aims_and_scope.html ). Therefore, this paper is suitable for publication in the Hess. We have revised the manuscript based on the comments. We hope this paper can be accepted for publication in the HESS. Reference: Awan, U. K. and Ismaeel, A.: A new technique to map groundwater recharge in irrigated areas using a SWAT model under changing climate, Journal of Hydrology, 519, 1368-1382, 2014. Falkemnark, M.: Coping with Water Scarcity under Rapid Population Growth: paper for the conference of SADC water ministries. Pretoria, 1995. Moriasi, D. N., Arnold, J. G., Van Liew, M. W., Bingner, R. L., Harmel, R. D., and Veith, T. L.: Model evaluation guidelines for systematic quantification of accuracy in watershed simulations, T Asabe, 50, 885-900, 2007. Troin, M. and Caya, D.: Evaluating the SWAT's snow hydrology over a Northern Quebec watershed, Hydrol Process, 28, 1858-1873, 2014.

Please also note the supplement to this comment:
http://www.hydrol-earth-syst-sci-discuss.net/hess-2016-241/hess-2016-241-AC1-supplement.zip

---

## Author Comment (AC2) · 12 Sep 2016

Dear Sir: We are very grateful for your insightful comments and suggestions, which have improved our manuscript. We have revised the manuscript based on your comments. We hope that it would be accepted for publication in HESS. Our responses are in blue. Thank you once again. Yours truly, Fulu Tao

1 Comment: However, I regret to say that neither the proposed methodology Response: Most of the previous studies have focused on the streamflow and groundwater that can be directly used for human activities. Since the concept of green and blue water was introduced by Falkenmark (1995), green/blue water research has become more and more diversified, especially after Falkenmark and Rockstrom (2006)

conceptualized a wider green-blue flows approach for water-resource planning and management. Many novel research methods have appeared as well. Recent studies have focused mainly on developing a concept or theoretical, developing simulation models and estimating quantities. However, County-level studies were rare. There are several studies about green and blue water at world, country, and basin scale. These studies were limited to regional scale and thus common results may not be derived for county-level assessment. County-level blue and green analysis is quite important and informative for water managers to formulate specific and suitable strategies. We admit that regional/basin studies can give us a big picture, but developing adaptive strategies for addressing the possible risks does need the local studies because all these strategies/schemes need to be implemented at level of county. In practice, there is an urgent need for decision makers to understand the green and blue water flow in each county. However, little previous researches especially focused on this scale and there is lack of a method and framework for the assessment of green and blue water flow at county-level, which was a priority focus of the research and a problem that needed an urgent attention for water management.Chinese government also provided many fund for addressing this problem. In this study, we proposed a framework and method for the assessment of green and blue water at the county level, combing SWAT hydrological model and statistical methods. Based on this method and framework, we assessed the green and blue water in the Heihe River Basin of China during 1980-2009. This study provides the reference for further studies on the county scale in similar regions and basin. Therefore, this paper is supposed to contribute greatly to the methodology that can assesse the county-level green and blue water in arid and semiarid basins. 2 Comment: The author present mainly an application of SWAT in combination with different standard statistical methods to assess trends in simulated green and blue water components. Response: In this study, we proposed a framework and method for the assessment of water resources including blue water, green water and total water. The Heihe River Basin is just the experimental site used to verify our methodology, where the county-level assessment of water resources due to climate and land use change

needs urgent analysis for the water planning and management. Our work is also not just a case study, but provides reference for further studies to estimate the water resources at county level. Therefore, this paper mainly proposed a new framework and method for the assessment of water resources at county level and then turn out the feasibility of this method by experiment. 3 Comment: The presentation of the model results is very brief (0.75 pages), the author mainly declare that their model is suited for the objective of their study and start straight away discussing the trends in simulated flows. Response: In the beginning, in view of the limitation of paper lenghth, the model results did not be presented and discussed in details. We have done many works during experiments which include yearly and monthly calibration and validation of hydrological model and comparing the simulated evapotranspiration (green water) with measured evapotranspiration by remote sensing technology. Based on this suggestion, in the revised version, we have stated the model results in 3.1 section in details. 4 Comment: Contrary to the author's statement, I found their results are not so convincing. For instance the authors evaluate their model on a monthly basis, regardless of the fact that river discharge reveals a strong seasonal pattern. In such a case, and as discussed by Schaefli and Gupta (2007), the null model is not the overall mean of the discharge, but the mean annual cycle of daily discharges. A proper evaluation implied either to benchmark the model against predicted deviations from the annual cycle, or to work at the daily scale. Response: Although Schaefli and Gupta (2007) discussed the Nash–Sutcliffe efficiency measure when reporting the results of a catchment modeling study, these conclusions from this paper can not be used to judge our model. Because this paper discussed the defect of Nash–Sutcliffe efficiency measure when we only use the NS to estimate the model. If we only used the NS to estimate model results, we will ignore the seasonal pattern of river discharge. In our work, we have considered the influence from seasonal pattern, because we used four indexes including the Nash-Sutcliffe efficiency coefficient (NS), coefficient, percent bias (PBIAS) and RMSE-observation standard deviation ratio (RSR) (Awan and Ismaeel, 2014; Moriasi et al., 2007; Troin and Caya, 2014) to estimate the model results. Moriasi and Arnold

(2007) developed comprehensive Standardization guidelines for model evaluation. According to this literature wrote, on the monthly scale, a model simulation is rated as good if 0.65 < NS < 0.75, 0.50 < RSR < 0.60 and ±10% < PBIAS < ±15%. A model simulation is judged as satisfactory if 0.50 < NS < 0.65, 0.60 < RSR < 0.70 and ±15% < PBIAS < ±25% (Moriasi et al., 2007). The simulation effect is by Moriasi and Arnold (2007) when RSR belongs to 0.60-0.70 and NS belongs to 0.50-0.65. Base on this, we did not present the results of uncertainty analysis on year scale. Base on this, we did not present the model evaluation results on the year scale. As a matter of fact, we evaluated the model on both of monthly and yearly basis during processes of experiment. And our model evaluation show that the hydrological performance is better on the year scale than that on the month scale. However, we added the content in estimate of yearly model results and stated in 3.1 section based on the suggestion. Meanwhile, in order to ensure convincing, we further assessed the accuracy of simulated actual evatranspiration by observed actual evapotranspiration from remote sensing observed for 2000, 2005, 2006, 2008 and 2009 year. We used relative error to estimate the results of simulated actual evapotranspiration. The comparison showed the SWAT model performance was good, with relative small error. The content was presented in 3.1 section. Therefore, the results are convincing. 5 Comment: Secondly, the authors compare simulated and annual ET totals to underpin that their model is well suited to discriminate green and blue water flows. Firstly, I wonder how the authors estimated ET annual totals, unfortunately the manuscript does not provide any information on the data sources which are used to drive and test the model. If their ET estimation is based on the long term water balance (P-Q), this cannot be regarded as independent assessment. Response: The actual evapotranspiration (ET) used to validation is observed actual evapotranspiration from remote sensing , rather than water balance (P-Q) . These data get Chinese official recognition and can be downloaded from West Data Center of China (WDCC, http://westdc.westgis.ac.cn/) . To ensure accuracy, we further assessed the accuracy of simulated actual evaportranspiration by observed actual evapotranspiration from remote sensing observed during 2000-2010

in the revised manuscript. 6 Comment: Secondly, one cannot conclude to have a model which reproduces green water flow dynamics well, without comparing the model against dynamic data. Response: Model calibration is the process of estimating model parameters by comparing model predictions (out-put) for a given set of assumed conditions with observed data for the same conditions. Model validation involves running a model using input parameters measured or determined during the calibration process. According to Refsgaard (1997), model validation is the process of demonstrating that a given site-specific model is capable of making "sufficiently accurate" simulations, although "sufficiently accurate" can vary based on project goals. If the performance of hydrological model we build is acceptable using the meansured data during calibration and validation periods, the model we build can be used to carry out the special goals. Our estimation results indicate the performance of model we build is good, so the build model can be applied into our study. Therefore, we have changed this sentence. Please see the 3.1 section in the revised manuscript. 7 Comment: The authors might consider either to largely enhance the scientific depth of their study or to submit their work to a more applied journal. Response: We have largely enhanced the scientific depth of our study based on your comments. We hope this paper can be accepted for publication in the HESS.

Reference: Awan, U. K. and Ismaeel, A.: A new technique to map groundwater recharge in irrigated areas using a SWAT model under changing climate, Journal of Hydrology, 519, 1368-1382, 2014. Falkemnark, M.: Coping with Water Scarcity under Rapid Population Growth: paper for the conference of SADC water ministries. Pretoria, 1995. Falkenmark, M. and Rockstrom, J.: The new blue and green water paradigm: Breaking new ground for water resources planning and management, J Water Res Pl-Asce, 132, 129-132, 2006. Moriasi, D. N., Arnold, J. G., Van Liew, M. W., Bingner, R. L., Harmel, R. D., and Veith, T. L.: Model evaluation guidelines for systematic quantification of accuracy in watershed simulations, T Asabe, 50, 885-900, 2007. Moriasi, D. N., Arnold, J. G., Van Liew, M. W., Bingner, R. L., Harmel, R. D., and Veith, T. L.: Model evaluation guidelines for systematic quantification of accuracy

in watershed simulations, T Asabe, 50, 885-900, 2007. Troin, M. and Caya, D.: Evaluating the SWAT's snow hydrology over a Northern Quebec watershed, Hydrol Process, 28, 1858-1873, 2014. Refsgaard, J. C.: Parameterisation, calibration and validation of distributed hydrological models, Journal of Hydrology, 198, 69-97, 1997.

Please also note the supplement to this comment:
http://www.hydrol-earth-syst-sci-discuss.net/hess-2016-241/hess-2016-241-AC2-supplement.zip